# pH Behavior of Polymer Complexes between Poly(carboxylic acids) and Poly(acrylamide derivatives) Using a Fluorescence Label Technique

**DOI:** 10.3390/polym11071196

**Published:** 2019-07-17

**Authors:** Yuriko Matsumura, Kaoru Iwai

**Affiliations:** 1Postgraduate School of Healthcare, Tokyo Healthcare University, 4-1-17 Higashi-Gotanda, Shinagawa-ku, Tokyo 141-8648, Japan; 2Professor Emeritus at Nara Women’s University, Kitauoya-Nishimachi, Nara 630-8506, Japan

**Keywords:** poly(acrylic acid), poly(methacrylic acid), poly(acrylamide derivatives), complexation, fluorescence label technique

## Abstract

In order to clarify the local environment during interpolymer complex formation between poly(carboxylic acids) and poly(acrylamide derivatives) with different *N*-substitutions, a fluorescence label technique was used. 3-(2-propenyl)-9-(4-*N*,*N*-dimethylaminophenyl) phenanthrene (VDP) was used as an intramolecular fluorescence probe. All polymers were synthesized by free radical polymerization. Interpolymer complexation was monitored by charge transfer emission from the VDP unit. Both of the poly(carboxylic acids) formed interpolymer complexes with poly(*N*,*N*-dimethylacrylamide) (polyDMAM). The micro-environments around the VDP unit in the acidic pH region for the poly(methacrylic acid) (polyMAAc) and polyDMAM mixed systems were more hydrophobic than those of the poly(acrylic acid) (polyAAc) and polyDMAM mixed systems, as the α-methyl group of the MAAc unit contributed to hydrophobicity around the polymer chain during hydrogen bond formation. This suggests that, when the poly(carboxylic acids) and poly(acrylamide derivatives) were mixed, with a subsequent decrease in the solution pH, a hydrogen bond was partially formed, following which the hydrophobicity of the micro-environment around the polymer chains was changed, resulting in the formation of interpolymer complexes. Moreover, the electron-donating ability of the carbonyl group in the poly(acrylamide derivatives) had an effect on complexation with poly(carboxylic acids).

## 1. Introduction

Poly(acrylic acid) (polyAAc) has been shown to form interpolymer complexes with poly(ethylene glycol) and poly(acrylamide) [1]. Interpolymer complexes have been recently intensively investigated in the field of developing new bio-compatible materials [2]. These interpolymer complexes are formed by hydrogen bonding between polymers containing hydrogen bond-donating groups and polymers containing hydrogen bond-accepting groups. Interpolymer complexes of polymer solutions or polymer blends have been widely investigated in the pharmaceutical field, especially in the study of drug delivery techniques [3]. Recently, polymer–polymer complexes between poly(carboxylic acids) and thermo-responsive polyacrylamides were investigated in an organic solvent [4]. However, the processes of interpolymer complex formation, including the local environment around the polymer chains, have not been studied much.

Fluorescence methods are powerful tools for investigating the conformational changes of polymers and micelle formation. This method has been widely used in the study of polymer–polymer interactions and micelle formation [5]. When a fluorescent probe is added into the interpolymer complex formation system, the environment around the fluorescent probe, located somewhere in the system, can be determined. However, information on the environment of the polymer chains cannot be obtained. In our previous paper, we reported the thermo-responsive behavior and micro-environments of poly(*N*-isopropylacrylamide) microgel particles labeled with the polarity-sensitive fluorescent molecule 3-(2-propenyl)-9-(4-*N*,*N*-dimethylaminophenyl)phenanthrene (VDP) (chemical structure shown in Figure 1) dispersed in water [6]. The VDP units inside the microgel particles became hydrophobic in conjunction with the phase transition of the microgel particles. Thus, information around the polymer chain where the VDP unit exists can be obtained by using VDP as an intrapolymer fluorescent probe.

In this paper, the fluorescent labeling method was used in order to clarify the local environment during interpolymer complex formation. Poly(acrylic acid) (polyAAc) and poly(methacrylic acid) (polyMAAc) were the poly(carboxylic acids) used, and three kinds of poly(acrylamide derivatives) with different *N*-substituents, poly(*N*,*N*-dimethylacrylamide) (polyDMAM), poly(*N*-ethyl-*N*-methylacrylamide) (polyEMAM), and poly(*N*,*N*-diethylacrylamide) (polyDEAM), were employed.

## 2. Materials and Methods

### 2.1. Materials

Acrylic acid (AAc) and methacrylic acid (MAAc) were purchased from Wako Pure Chemicals (Osaka, Japan) and purified by vacuum distillation. *N*,*N*-Dimethylacrylamide (DMAM) was purchased from Tokyo Kasei Kogyo (Tokyo, Japan) and purified by vacuum distillation. α,α ‘-azobisisobutyronitrile (AIBN) was purchased from Wako Pure Chemicals (Osaka, Japan) and recrystallized twice from methanol. *N*-ethyl-*N*-methylacrylamide (EMAM) was prepared by the acrylation of *N*-ethyl-*N*-methylamine with acryloyl chloride. *N*,*N*-diethylacrylamide (DEAM) was prepared by the acrylation of *N*,*N*-diethylamine with acryloyl chloride. All other reagents were of guaranteed reagent grade and used without further purification. The fluorescent probe monomer VDP and its monomer unit model compound 3-ethyl-9-(4-*N*,*N*-dimethylaminophenyl) phenanthrene (EtDP) (chemical structure shown in Figure 1) were used as previously prepared [6].

### 2.2. Synthesis of VDP-Labeled Polymers

All polymers were synthesized by radical polymerization, using AIBN as an initiator, as reported previously [7]. PolyAAc and poly(VDP-*co*-AAc) were synthesized as follows: AAc (1 mol/L), fluorescent monomer VDP (0 or 1 mmol/L), and AIBN (5 mmol/L) were dissolved in methanol. The solution was degassed by the freeze-pump-thaw method, heated to 60 °C for 6 h, and then cooled to room temperature. The reaction mixture was poured into a 20 times amount of ethyl acetate. The obtained polymer was purified by re-precipitation, using methanol as a solvent and ethyl acetate as a precipitant. The obtained polymers were dried under vacuum for 24 h. The VDP unit contents in the polymers were determined from the absorbance of methanol solutions (8–10 mg/10 mL), as compared to EtDP (ε = 16,100 mol^−1^·L·cm^−1^ at 314 nm) [6] as a model compound. The absorption spectra were measured on a HITACHI U-3200 spectrophotometer. PolyMAAc, poly(VDP-*co*-MAAc), polyDMAM, poly(VDP-*co*-DMAM), poly(VDP-*co*-EMAM), and poly(VDP-*co*-DEAM) were prepared by a similar procedure, as shown in Table 1.

### 2.3. Fluorescence Measurements

The stock polymer solutions were separately prepared at a concentration of 0.01 w/v% using distilled water. The polymer solutions were mixed with equal amounts of polymer solution and distilled water and bubbled with nitrogen gas at 25 °C for 30 min. The pH of the solution was changed by adding 0.02 mol/L of NaOH or 0.02 mol/L of HCl solution under a nitrogen atmosphere and measured using a Horiba D-13 pH meter. The fluorescence spectra of the polymer solutions were measured at 25 °C under a nitrogen atmosphere using a Hitachi F-2500 fluorescence spectrophotometer. The excitation wavelength was set at 320 nm to excite the VDP unit in the polymers.

### 2.4. Fraction of Dissociated Carboxyl Group (α) of Poly(Carboxylic Acids)

A pH titration was performed by adding 0.02 mol/L NaOH solution to 50 mL of polymer solution (0.005 w/v%). The fraction of dissociated carboxylic group of the poly(carboxylic acids) (α) was calculated using the equation:
(1)
α = (*C*_a_ + *C*_H+_ − *C*_OH-_)/*C*_p_
where *C*_a_, *C*_H+_, *C*_OH-_, and *C*_p_ denote the concentrations of sodium ions (mol/L), protons (mol/L), hydroxide ions (mol/L), and the initial polymer (unit mol/L), respectively.

## 3. Results and Discussion

### 3.1. pH Responsive Behavior of VDP-Labeled Polymers

Typical fluorescence spectra of poly(VDP-*co*-AAc) and poly(VDP-*co*-DMAM) solutions at various pH are shown in Figure 2. All of the VDP-labeled polymer solutions exhibited emission at 340–400 nm from protonation of the nitrogen atom of the VDP units, and a broad intramolecular charge transfer (ICT) emission (400–600 nm) from the VDP units. The wavelength at the maximum intensity (λ_max_) for ICT emission at pH 4.3 of the poly(VDP-*co*-AAc) solution shifted to a shorter wavelength, compared with that at pH 10.2. On the other hand, the λ_max_ values of the poly(VDP-*co*-DMAM) solution were not changed with varying pH. This blue-shift of the λ_max_ value reflects an increase of hydrophobicity around the VDP units, as reported previously [6]. When the pH of the poly(carboxylic acids) solutions was changed from acidic to alkaline, the carboxylic groups dissociate and electrostatic repulsion occurred between the carboxylate anions. The conformation of the polymer chains was changed from a hypercoiled structure to a water-swollen state with a decrease in the solution pH [8,9]. As the λ_max_ value for poly(VDP-*co*-DMAM) was not affected by the pH of the polymer solution, the VDP units in the polymers could sense and report this formation of the polymer chains in response to the pH.

As discussed in detail, the fraction of dissociated carboxyl group (α) and the λ_max_ values for poly(VDP-*co*-AAc) and poly(VDP-*co*-MAAc) are plotted against pH in Figure 3. The dissociation constants for poly(VDP-*co*-AAc) and poly(VDP-*co*-MAAc) were 7.3 and 8.1, respectively, meaning that poly(VDP-*co*-MAAc) was a weaker acid than poly(VDP-*co*-AAc). Poly(VDP-*co*-AAc) showed a one-step dissociation, whereas poly(VDP-*co*-MAAc) showed a two-step dissociation. It has been reported that polyMAAc presents small-scale rearrangements in structure around the acidic region, rather than a large-scale expansion, which is then followed by a macroscopic change in dimension at the neutralization point [8]. The λ_max_ value gradually blue-shifted with decreasing pH for both polymer solutions. The λ_max_ values were almost constant at 490 nm, with above 50% dissociation of the carboxyl group for both poly(carboxylic acids), and was blue-shifted with a decrease in the dissociation degree, meaning that the hydrophobicity around the VDP unit increased. In the acidic region, the λ_max_ values for poly(VDP-*co*-MAAc) (420 nm at pH 3.3) were smaller than those for poly(VDP-*co*-AAc) (453 nm at pH 3.3). The λ_max_ values were linearly correlated with the solvent polarity (ε), and the following equation was obtained by least-squares analysis (where the r value is the correlation coefficient), as reported previously [6]:
(2)
λ_max_ (nm) = 0.9859 ε + 418.24 (r = 0.998).


The micro-environmental polarity near the VDP units in the polymers was estimated from the observed λ_max_ value using Equation (2). The estimated ε value for poly(VDP-*co*-MAAc) at pH 3.3 was <8, which was smaller than that for poly(VDP-*co*-AAc) (estimated ε = 35). The estimated ε values were 73 at pH 9 for both polymers. The MAAc unit has an α-methyl group, which affects the polymer conformation where the hydrogen bond is formed between the carboxylate groups. This means that the hydrophobicity around the VDP units in poly(VDP-*co*-MAAc) is large, compared to that in poly(VDP-*co*-AAc). On the other hand, in the alkaline pH region, where the electric repulsion causes the hypercoiled form, the VDP units sense a similar hydrophilic environment.

### 3.2. Complexation Between PolyDMAM With Poly(Carboxylic Acids)

Polymers containing hydrogen-donating groups were mixed with polymers containing hydrogen-accepting groups, resulting in the formation of interpolymer complexes, as shown in Figure 4. When poly(VDP-*co*-AAc) was mixed with polyDMAM, the scattering intensity increased in the acidic region (under pH 4), as shown in Figure 5a. The increase in scattering intensity means that the turbidity of the solution increased. This increase in scattering intensity was also observed for the polyAAc and poly(VDP-*co*-DMAM) mixed system. These phenomena were also observed for the polyMAAc and polyDMAM mixed systems, as shown in Figure 5b. Poly(carboxylic acids) can form complexes with polyDMAM by hydrogen bonding between the H atom of the carboxylic acid in the AAc unit and the O atom of the carbonyl group in the DMAM unit [10,11]. This hydrogen bond formation results in interpolymer complex formation and an increased turbidity of the mixed solution.

When poly(VDP-*co*-DMAM) was mixed with polyAAc or polyMAAc, the fluorescent λ_max_ value blue-shifted with a decrease in the solution pH, at around pH 6, as shown in Figure 5c,d. As interpolymer complexation caused an increase in turbidity, this blue-shift means that the hydrophobicity around the VDP unit increased, as discussed above, and that interpolymer interaction occurred even above pH values where the interpolymer complexations were occurred. Thus, these findings suggest that, with decreasing solution pH, interpolymer hydrogen bonds were partially formed, following which the interpolymer complex formed. The fluorescent λ_max_ profile for the poly(VDP-*co*-DMAM) and PAAc mixed system was different, compared to that of the polyDMAM and poly(VDP-*co*-AAc) mixed system. The fluorescent λ_max_ values for poly(VDP-*co*-AAc) were smaller than those of poly(VDP-*co*-DMAM) in the polymer mixed systems, in the range of pH 4–6. This blue-shift was also observed under pH 4, where the interpolymer complex had formed. On the other hand, the fluorescent λ_max_ value for poly(VDP-*co*-MAAc) was of a shorter wavelength than that for poly(VDP-*co*-DMAM), in the range of pH 6.5–7, and was the same under pH 6.5. The fluorescent λ_max_ profiles differed with the poly(carboxylic acids) used. The micro-environment around the VDP unit in polyMAAc was more hydrophobic than polyAAc in the acidic pH region, as the α-methyl group in the MAAc unit affected the hydrophobicity around the polymer chain during hydrogen bond formation.

### 3.3. Complexation Between PolyAAc and VDP-Labeled Poly(Acrylamide Derivatives)

Drastic changes in the micro-environments around the polymer chains were observed during complex formation when polyDMAM was labeled with the VDP unit. In the case of complex formation between polyAAc and poly(acrylamide derivatives), the VDP unit labeled the poly(acrylamide derivatives). When polyAAc was mixed with poly(VDP-*co*-acrylamide derivatives), the scattering intensities at 320 nm were increased for all polymer mixed solutions with decreased pH, where the threshold pHs were 4.2, 4.8, and 4.9 for poly(VDP-*co*-DMAM), poly(VDP-*co*-DEAM), and poly(VDP-*co*-EMAM), respectively (Figure 6a). Complex formation was observed for all poly(acrylamide derivatives) and polyAAc mixed systems. On the other hand, the fluorescence λ_max_ values were shifted to short wavelengths at a pH of 6.3, 5.7, and 5.3 for poly(VDP-*co*-DEAM), poly(VDP-*co*-EMAM), and poly(VDP-*co*-DMAM), respectively (Figure 6b). These pH values were higher than those of the interpolymer complex formations. This suggests that, when the poly(carboxylic acids) and poly(acrylamide derivatives) were mixed and then the pH was decreased, a hydrogen bond was partially formed and, when the micro-environment around the polymer chains became hydrophobic, the formation of the interpolymer complexes resulted. It has been reported that complexation between polyMAAc and polyDEAM occurs due to the formation of hydrogen bonds, and a ladder model was proposed at low pH [12]. In the case of our study, this proposed ladder model was also applicable to the structures of the polyAAc and poly(acrylamide derivative) complexes. The N atom is an electron donor, which will increase the electron density on the carbonyl O. Thus, the carbonyl O in poly(acrylamide derivatives) may increase hydrogen bond probability. The electron-donating ability of the N side-chain has been explained in terms of the oxidation potentials of amines with different N-substituents. The oxidation potentials for trimethylamide, dimethyl-ethylamine, and triethylamine were 0.76, 0.74, and 0.69 eV (vs. SCE in Britton–Robinson Buffer solution at pH 11.9), respectively [13]. The electron-donating ability of the ethyl group is larger than that of the methyl group. The electron-donating ability of the carbonyl group in poly(acrylamide derivatives) is affected by the electron-donating ability of the amine group. As the polymers with higher electron-donating ability had a strong proton acceptor, poly(VDP-*co*-DEAM) formed an interpolymer complex with polyAAc at the highest pH region of the three poly(acrylamide derivatives). The micro-environment around the VDP unit in the acidic pH region was affected not only by the proton-accepting ability, but also by the hydrophobicity of the poly(acrylamide derivatives) used.

## 4. Conclusions

In this paper, the interpolymer complexes between two kinds of poly(carboxylic acids) and three kinds of poly(acrylamide derivatives) with different *N*-substituent groups were investigated. Both of the poly(carboxylic acids) formed interpolymer complexes with polyDMAM. The micro-environments around the VDP unit in the acidic pH region for the polyMAAc and polyDMAM mixed systems were more hydrophobic than those in the polyAAc and polyDMAM mixed systems, as the α-methyl group of the MAAc unit affected the hydrophobicity around the polymer chain during hydrogen bond formation. This suggests that when the poly(carboxylic acids) and poly(acrylamide derivatives) were mixed and then, the solution pH was decreased, a hydrogen bond was partially formed, following which the micro-environment around the polymer chains changed in hydrophobicity, resulting in the formation of interpolymer complexes. The electron-donating ability of the carbonyl group in the poly(acrylamide derivatives) affected the complexation with polyAAc.

## Figures and Tables

**Figure 1 polymers-11-01196-f001:**
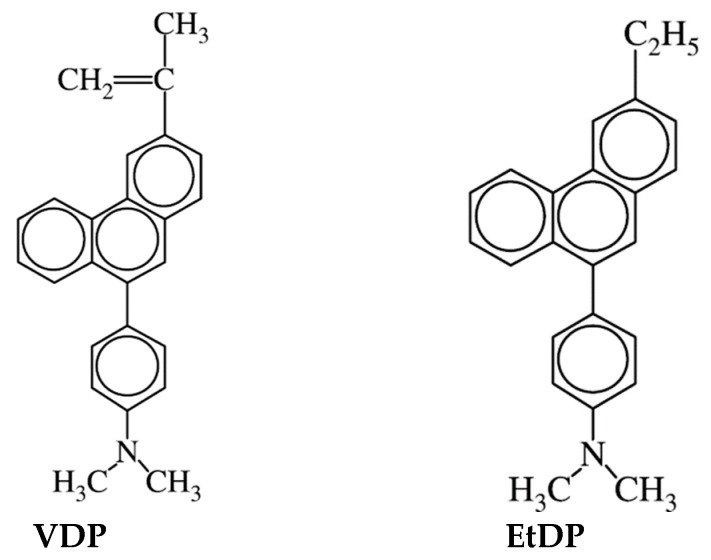
Chemical structures of 3-(2-propenyl)-9-(4-*N*,*N*-dimethylaminophenyl)phenanthrene (VDP) and 3-ethyl-9-(4-*N*,*N*-dimethylaminophenyl) phenanthrene (EtDP).

**Figure 2 polymers-11-01196-f002:**
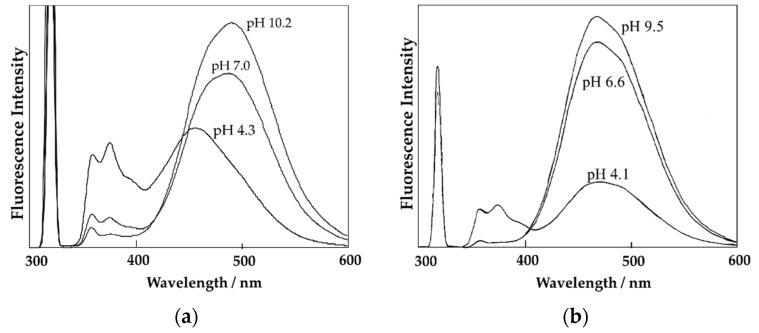
Fluorescence spectra of the VDP-labeled polymer solutions: (**a**) poly(VDP-*co*-AAc) and (**b**) poly(VDP-*co*-DMAM).

**Figure 3 polymers-11-01196-f003:**
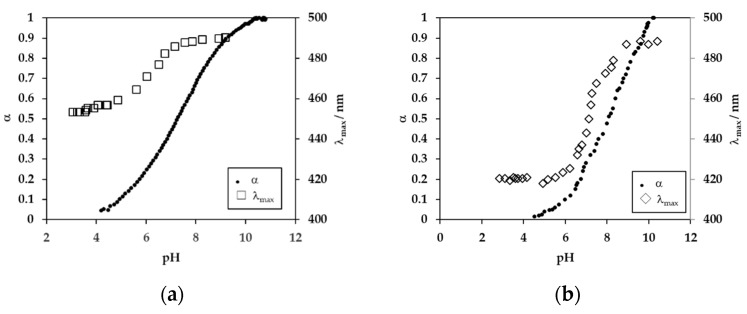
The fraction of dissociated carboxylic group (α) and fluorescence λ_max_ values for 0.005 w/v% poly(VDP-*co*-carboxylic acids) solutions: (**a**) poly(VDP-*co*-AAc) and (**b**) poly(VDP-*co*-MAAc).

**Figure 4 polymers-11-01196-f004:**
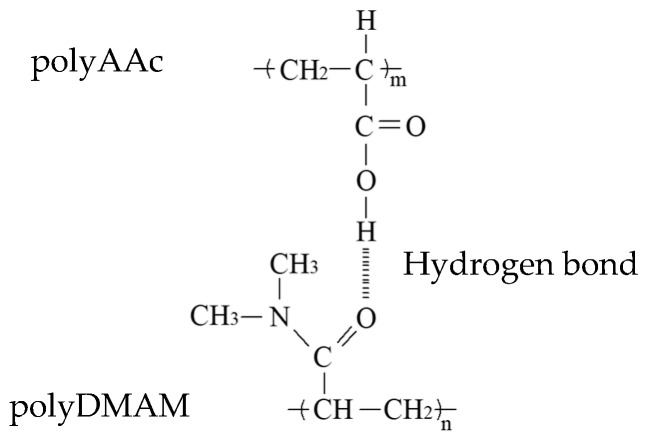
The polymer complexation of poly(acrylic acid) (polyAAc) with poly(*N*,*N*-dimethylacrylamide) (polyDMAM) by hydrogen bonding.

**Figure 5 polymers-11-01196-f005:**
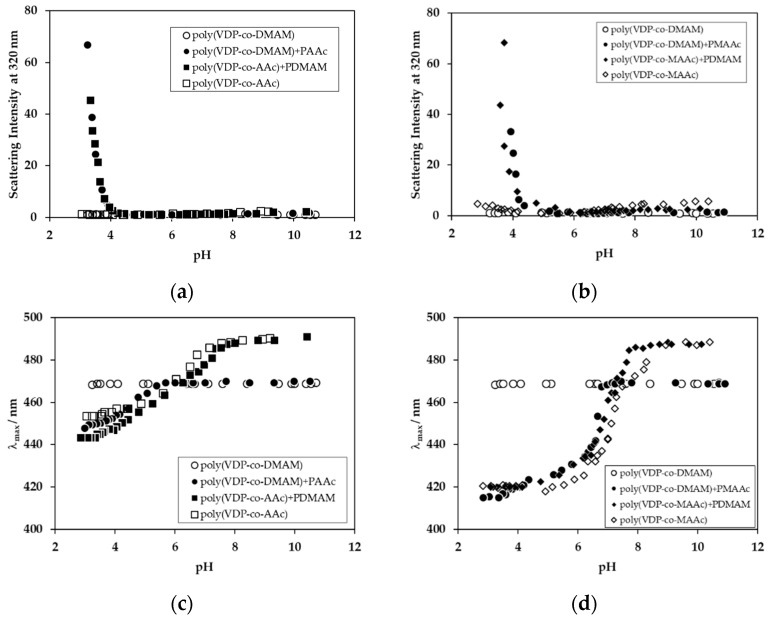
Scattering intensity at 320 nm (**a**,**b**) and fluorescence λ_max_ values (**c**,**d**) as a function of pH: PolyDMAM and polyAAc mixed systems (**a**,**c**) and polyDMAM and poly(methacrylic acid) (polyMAAc) mixed systems (**b**,**d**).

**Figure 6 polymers-11-01196-f006:**
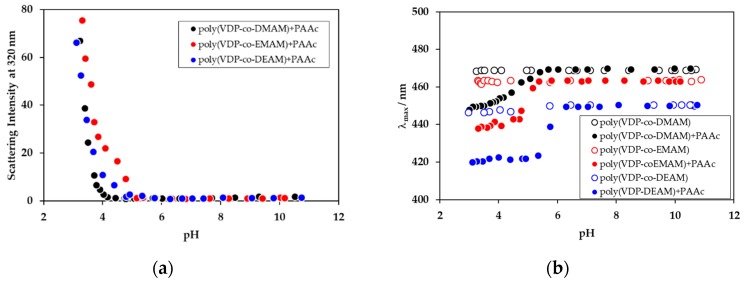
Scattering intensity at 320 nm and fluorescence λ_max_ values of the mixed solutions of poly(VDP-*co*-acrylamide derivatives) and polyAAc, as a function of pH.

**Table 1 polymers-11-01196-t001:** Recipes for polymerizations ^1^.

Polymer	Solvent	Precipitant	Reaction Time (h)	mol% of VDP Monomer		Yield (%)
				In Feed	In Polymer	
PolyAAc	methanol	ethyl acetate	6	0	0	72.7
Poly(VDP-*co*-AAc)	methanol	ethyl acetate	6	0.1	0.12	74.1
PolyMAAc	methanol	diethyl ether	6	0	0	68.0
Poly(VDP-*co*-MAAc)	methanol	diethyl ether	6	0.1	0.15	61.0
PolyDMAM	benzene	*n*-hexane	1	0	0	52.9
Poly(VDP-*co*-DMAM)	benzene	*n*-hexane	1	0.1	0.17	50.5
Poly(VDP-*co*-EMAM) ^2^	benzene	*n*-hexane	1	0.1	0.14	47.4
Poly(VDP-*co*-DEAM)	benzene	*n*-hexane	1	0.1	0.15	57.6

^1^ AIBN = 5 mmol/L, ∑ monomer = 1 mol/L at 60 °C. ^2^ ∑ monomer = 0.5 mol/L.

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
