# Peer review of "pH Behavior of Polymer Complexes between Poly(carboxylic acids) and Poly(acrylamide derivatives) Using a Fluorescence Label Technique"

_polymers, 2019, doi:10.3390/polym11071196_

Reviewer 1 Report

I recommend to publish this paper for Polymers. This paper includes interesting tool of fluorescent method for studying microenvironment around polymer chain, especially due to interpolymer complex formation.  

Here, I hope to mention some additional information as follows:

In Table 1, the yield of polyDMAM is 92.9%, however, that of poly(VDP-co-DMAM) is 50.5%. Why does difference occur?  Are the molecular weights of these polymers equal? If the molecular weight of these polymers are fairly different, the mechanism of interpolymer complex formation must be different.

Typing error: Line 190 last word thant (than) 

Author Response

Point 1: In Table 1, the yield of polyDMAM is 92.9%, however, that of poly(VDP-co-DMAM) is 50.5%. Why does difference occur?  Are the molecular weights of these polymers equal? If the molecular weight of these polymers are fairly different, the mechanism of interpolymer complex formation must be different.

Response 1: I checked all data in our manuscript. I made a mistype here. The correct yield of polyDMAM is 52.5%, not 92.5%. I correct the value in Table 1.

Point 2:Typing error: Line 190 last word thant (than) .

Response 2:Thank you very much for your comment. English editing was performed using MDPI service.

Reviewer 2 Report

The article (Ref.: polymers-541633) presents the complexation between poly(carboxylic acids) and poly(acrylamide derivatives) at different pH. 3-(2-propenyl)-9-(4-N,N-dimethylaminophenyl)phenanthrene (VDP) is used as a fluorescent label to study polymer complexation. The article is well-written and understandable, though there are a few English language and grammatical errors. Nevertheless, I may suggest some points to improve the manuscript.

Authors should give a general schematic of the polymer complexation mechanism, for instance, between poly(acrylic acid) and poly(N, N-dimethyl acrylamide). This will improve our understanding of the underlying interactions between polymers.

Authors state, "the electron donating ability of carbonyl group 24 in the poly(acrylamide derivatives) affected to the complexation with poly(carboxylic acids)." 
The carbonyl group is usually 'electron withdrawing', not an electron donating group. In acrylamide derivatives, e.g. N, N-dimethyl acrylamide, *N is electron donating* that will increase electron density on carbonyl O. Thus, carbonyl O in polyacrylamide derivatives may increase hydrogen-bonding probability. A schematic of the polymer complexation mechanism may describe the same in a better way.

Author Response

Point 1: Authors should give a general schematic of the polymer complexation mechanism, for instance, between poly(acrylic acid) and poly(N,N-dimethyl acrylamide). This will improve our understanding of the underlying interactions between polymers.

ʉ۬

Response 1: Thank you very much for your comment. We added the general scheme at line 150 and line 158 for the revised manuscript. And the sentences at line 198 to line 200 in the revised manuscript was changed.

Point 2:The article is well-written and understandable, though there are a few English language and grammatical errors.

Response 2:English editing was performed using MDPI service.
